# WHOLE-BRAIN CONNECTOMIC GRAPH NEURAL NETWORK ENABLES WHOLE-BODY LOCOMOTION CONTROL IN DROSOPHILA

## ABSTRACT

Whole-brain connectome provides a structural blueprint for linking neural circuits to behavior, yet its application to embodied control remains largely unexplored. We introduce the fly-connectomic Graph Neural Network (flyGNN), a reinforcement learning controller whose architecture is instantiated directly from a complete adult Drosophila connectome. Our flyGNN models the connectome as a directed message-passing graph, partitioned into afferent, intrinsic, and efferent pathways that structure information flow from sensory inputs to motor outputs. Integrated with a dynamically controllable biomechanical model of Drosophila, flyGNN achieves stable control across diverse locomotion tasks, including gait initiation, walking, turning, and flight, without task-specific architectural tuning. These results demonstrate that whole-brain connectivity can directly support embodied reinforcement learning, establishing a new paradigm for connectome-based control algorithms.

## 1 INTRODUCTION

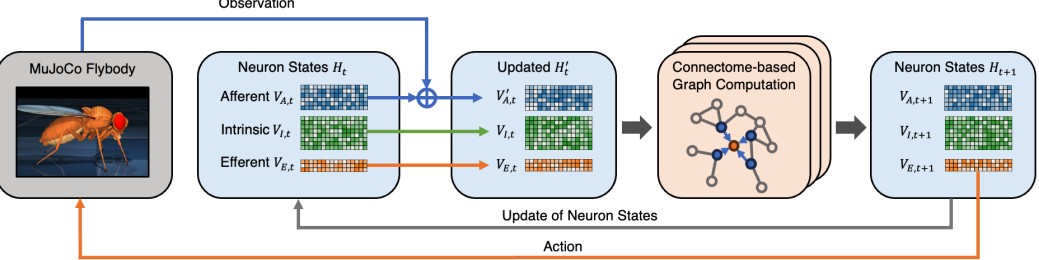

Figure 1: **Overview of the flyGNN enabled whole-body locomotion control framework.** Observations are mapped into afferent neuron states through dimensional transformation. Neural states are then propagated through the connectome-based graph computation module, where update rules are directly constrained by the FlyWire connectome. The updated efferent neuron states are converted into actions to drive the whole-body locomotion of an embodied drosophila model in MuJoCo. (An anonymous webpage with demonstration videos and source code is available at: https://sites.google.com/view/flygnn).

Understanding how neural circuits give rise to behavior is a long-standing challenge shared by neuroscience and artificial intelligence. Recent advances in whole-brain connectomics have provided open-source neuronal wiring diagrams of the adult Drosophila brain at synaptic resolution (Dorkenwald et al., 2024). These resources enable the possibility of linking complete brain structure to the sensorimotor control of a physical body. Yet, a fundamental challenge remains: how can static connectomes be transformed into dynamic, functional models that reproduce the intricate and adaptive motor behaviors of animals? Answering this question requires bridging two active lines of research: (i) mechanistic modeling that leverages whole-brain connectivity, and (ii) learning frameworks that generate high-dimensional whole-body movements.

On the control side, reinforcement learning has produced controllers that can drive humanoids, quadrupeds, and musculoskeletal agents to achieve challenging tasks. However, these policies typically rely on generic multilayer perceptrons (MLPs) or hand-crafted modules. While these architectures are useful, they bear little relevance to the biological structure of the brain, making it difficult to align them with real nervous systems. On the connectomics side, models constrained by circuit anatomy have provided insights into sensory and premotor computations (Azevedo et al., 2024), yet most efforts remain limited to specific subsystems and simplified behaviors. What is missing is a whole-brain, embodied approach that respects the connectivity blueprint and operate in a closed-loop physical environment.

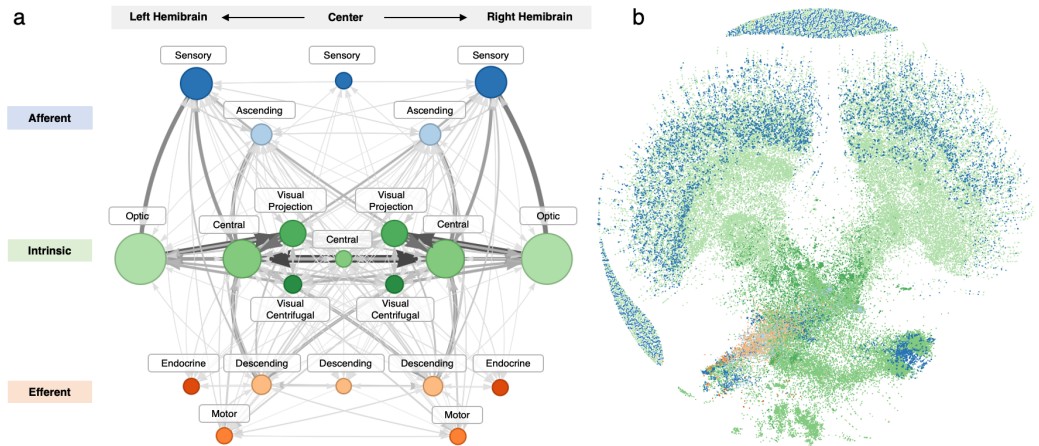

Figure 2: **Structure of the fly-connectomic Graph Neural Network.** **(a)** Aggregated synapse graph of the fly connectome, grouped into afferent (blue), intrinsic (green), and efferent (orange) sets across left hemibrain, central, and right hemibrain compartments. Node sizes reflect the number of neurons in each group, and arrows indicate the direction and relative strength of connectivity. **(b)** Force-directed graph layout (Kobourov, 2012) of the same neural network. The spatial layout reveals hemispheric symmetry and functional clustering.

Here, we bridge these domains by introducing the fly-connectomic Graph Neural Network (fly-GNN), a controller whose architecture is directly adapted from the Drosophila whole-brain connectome. We treat the connectome as an unweighted, directed graph and partition nodes into afferent, intrinsic, and efferent sets following the information flow directions. At each control step, flyGNN takes the sensory signals as input through afferent pathways, propagates the information through the brain-wide graph via message-passing, and outputs motor actions through the efferent pathways to actuate the body.

We evaluate flyGNN on a physics-based Drosophila model (Vaxenburg et al., 2025), demonstrating that a single connectome-grounded policy class supports multiple locomotor behaviors, including gait initiation, straight walking, turning, and flight. Beyond behavioral performance, we analyze neural representations with low-dimensional embeddings and the large-scale visualizations reveal emergent functional segregation across sensory, central, and motor populations. This differentiation emerges solely from the connectome topology without biological priors, suggesting that structural wiring may induce functional specialization.

In this work, we introduce fly-connectomic Graph Neural Network (flyGNN), a neural network controller whose architecture is directly adapted from a Drosophila whole-brain connectome. We show that this connectome-structured network can drive diverse locomotor behaviors in a physics-based fly simulation, including gait initiation, walking, turning, and flight. By demonstrating neural control of movement directly from the connectome, this work establishes a framework for studying how whole-brain neural architectures support whole-body movement behavior, advancing both neuroscience and embodied artificial intelligence.

## 2 RELATED WORK

**Connectomics-based neural network modeling.** Advances in connectomics have deepened our understanding of circuit-level organization in central nervous systems of animal models. In particular, the FlyWire project provides a whole-brain reconstruction of Drosophila at synaptic resolution, offering the structural basis for modeling complete neural dynamics (Dorkenwald et al., 2024; Schlegel et al., 2024; Zheng et al., 2018). Prior works have highlighted the explanatory power of such data in restricted domains: ventral nerve cord reconstructions revealed leg–wing coordination circuits (Azevedo et al., 2024; Lesser et al., 2024), and models of motor neurons have been applied to feeding or grooming behaviors (Shiu et al., 2024). Connectome-constrained networks have also been used to predict neural activity in the visual system (Lappalainen et al., 2024). However, these approaches often focus on specific subsystems or tasks, leaving open the question whether whole-brain connectivity can generate realistic control of embodied locomotion behaviors.

**Embodied movement control.** In parallel, embodied intelligence research has advanced locomotion in simulated humanoids (Kumar et al., 2021; Cheng et al., 2024), quadrupeds (Ding et al., 2021), and musculoskeletal agents (He et al., 2024; Wei et al., 2025) using reinforcement learning. In the Drosophila domain, physics-based models such as NeuroMechFly (Lobato-Rios et al., 2022; Wang-Chen et al., 2024) and flybody (Vaxenburg et al., 2025) have enabled detailed simulations of walking and flight in MuJoCo (Todorov et al., 2012). Yet, controllers for these systems are typically built from generic MLPs or manually designed central pattern generators, lacking direct biological grounding. This limits both interpretability and the ability to connect neural structure to behavior. Our work differs by directly embedding the connectome into the controller architecture, combining embodied simulation with structural priors to study both performance and neural representation.

## 3 METHOD

### 3.1 CONNECTOME-STRUCTURED NEURAL NETWORK ARCHITECTURE

We consider the problem of embodied sensorimotor control for a virtual fruit fly agent interacting with a physics-based environment provided by flybody. Let the state at time step $t$ be denoted by $s_t \in \mathcal{S}$. The agent receives an observation $x_t \in \mathbb{R}^{d_{\text{in}}}$, which corresponds to a set of processed features including proprioceptive and exteroceptive signals during movement and environmental interaction. Based on this input, the neural controller produces an action $a_t \in \mathbb{R}^{d_{\text{out}}}$, representing the motor outputs that drive the flybody model to perform locomotion behaviors.

The controller is parameterized by a graph neural network whose architecture follows the anatomical connectivity of the Drosophila brain. Specifically, the connectome is implemented as a directed graph $G = (V, E)$, where each node $v \in V$ corresponds to a neuron and each directed edge $(u, v) \in E$ indicates the existence of a synaptic connection from neuron $u$ to neuron $v$. We simplified further biological details (such as neurotransmitter types, synapse counts, or cell morphology) and modeled the connectome to capture the existence and direction of synaptic connections.

Following FlyWire's classification of flow types, we partitioned the nodes into three disjoint sets:

- **Afferent neurons**: $V_a \subset V$, which receive external sensory inputs,
- **Intrinsic neurons**: $V_i \subset V$, which mediate signals within the network, and
- **Efferent neurons**: $V_e \subset V$, which produce motor outputs to the body model.

Our flyGNN architecture implements the complete Drosophila connectome through a graph neural network where each of the 139,246 neurons (comprising 19,262 afferent, 118,496 intrinsic, and 1,488 efferent neurons) is represented by node embeddings. The architecture's biological fidelity and substantial capacity support complex locomotion capabilities while maintaining the structure of the connectome.

Thus, $V = V_a \cup V_i \cup V_e$ with $V_a \cap V_i \cap V_e = \emptyset$. Each neuron $v \in V$ is associated with a latent state vector $h_v \in \mathbb{R}^C$, and we collected all neuron states at time $t$ as a matrix:

$$H_t \in \mathbb{R}^{|V| \times C} \tag{1}$$

At each step, the sensory input $x_t$ was first embedded into afferent states by an encoder layer $\text{Enc}_\theta$:

$$z_t = \text{Enc}_\theta(x_t) \in \mathbb{R}^{|V_a| \times C} \tag{2}$$

The afferent states were then updated by a gated mechanism that combined the encoded sensory features and the previous afferent states:

$$h'_a = \tanh\Big(W_g\,[\,z_t \,\|\, H_t[V_a]\,] + b_g\Big), \quad a \in V_a, \tag{3}$$

where $[\,\cdot\|\cdot\,]$ denotes concatenation and $W_g, b_g$ are learnable parameters. This update ensures that sensory information is integrated with the existing neural states before message-passing.

After afferent updates, the complete node states $H_t$ were propagated through the connectome graph by a message-passing operator $\text{MP}(\cdot)$. In the simplest case of a Graph Convolutional Network (GCN) (Kipf & Welling, 2017), the update rule could be formulated as the following:

$$H^{(l+1)} = \sigma\Big(\tilde{D}^{-\frac{1}{2}}\tilde{A}\tilde{D}^{-\frac{1}{2}}H^{(l)}W^{(l)}\Big) \tag{4}$$

where $\tilde{A} = A + I$ is the adjacency matrix with self-loops, $\tilde{D}$ is the diagonal degree matrix, $W^{(l)}$ is a learnable weight matrix, and $\sigma(\cdot)$ is a nonlinear activation. This operator propagated neural information along the edges of the connectome, producing updated states for both intrinsic and efferent neurons. In practice, the message-passing module can be instantiated not only as a GCN but also with more expressive architectures such as GraphSAGE (Hamilton et al., 2018), GAT (Veličković et al., 2018), or PNA (Corso et al., 2020), which offered improved performance and scalability (See Appendix C for comparative experiments).

The updated efferent states $H_t[V_e]$ were flattened and passed through a decoder layer $\text{Dec}_\phi$:

$$a_t = \text{Dec}_\phi\Big(H_t[V_e]\Big) \tag{5}$$

This step maps the neural features of the efferent neurons directly into continuous motor outputs, denoted as $a_t$. These outputs served as motor commands to actuate the flybody, a biomechanical model of Drosophila implemented in MuJoCo (Vaxenburg et al., 2025). The agent then updated the physical state of the fly and produced the next sensory observation $x_{t+1}$, making the sensorimotor in a closed-loop manner. Over a trajectory $\tau = (x_0, a_0, x_1, \dots, x_T)$, the objective of the fly agent was to generate actions that realize stable and efficient locomotion.

In summary, Algorithm 1 outlines the complete forward computation of our neural control policy which follows the anatomical connectivity of the Drosophila brain - from sensory input through afferent gating, message-passing across the connectome, to efferent decoding into motor actions.

---

**Algorithm 1:** fly-connectomic Graph Neural Network (flyGNN)

---

**Input:** Sensory input $x_t$, connectome graph $G = (V, E)$ with node partitions $V_a$, $V_i$, $V_e$;
        encoder Enc; gate $(W_g, b_g)$; message-passing MP; decoder Dec
**Output:** Motor output $a_t$
**for** *each time step $t$* **do**
    $z_t \leftarrow \text{Enc}(x_t)$ ;
    $H_t[V_a] \leftarrow \tanh\big(W_g[\,z_t, H_t[V_a]\,] + b_g\big)$ ;
    $H_{t+1} \leftarrow \text{MP}(H_t, E)$ ;
    $a_t \leftarrow \text{Dec}(H_{t+1}[V_e])$ ;
    Apply $a_t$ to MuJoCo to obtain $x_{t+1}$ ;
**end**

---

## 3.2 TRAINING PIPELINE

Our training pipeline consisted of two stages: first, we initialized the connectome-based policy using imitation learning from expert trajectories, and second, we fine-tuned the model with reinforcement

learning to directly optimize for task rewards. This two-stage design leveraged demonstration data for rapid initialization while preserving the capability for adaptive policy improvement.

To provide an initial policy, we collected expert trajectories by rolling out an MLP-based policy for the flybody which was originally trained with imitation learning to generate high-quality demonstrations of locomotion. We used these trajectories to train our connectome-based model by imitating the expert's action distributions.

Specifically, the policy predicted Gaussian parameters $(\mu_s, \sigma_s)$ given the same observations as the expert, and was optimized to minimize a loss combining Kullback–Leibler divergence with an annealed mean squared error (MSE) regularizer:

$$\mathcal{L}_t = D_{\mathrm{KL}}\big(\mathcal{N}(\mu_t, \sigma_t^2) \,\|\, \mathcal{N}(\mu_s, \sigma_s^2)\big) + \lambda(t)\Big(\|\mu_s - \mu_t\|_2^2 + \alpha \,\|\log \sigma_s - \log \sigma_t\|_2^2\Big) \quad (6)$$

where $\alpha$ is a constant to balance the scale of $\mu_s$ and $\log \sigma_s$, and $\lambda(t)$ decreases during training so that distributional matching dominates in later stages. This procedure initialized the model with stable behaviors for walking and flight tasks.

After initialization, we fine-tuned the connectome-structured policy using Proximal Policy Optimization (PPO) to enable direct learning from rewards. For value estimation, we used a simple MLP as the value network. The environments in MuJoCo were adapted into gym-like interfaces with parallel rollouts to increase throughput, and distributed training with Distributed Data Parallel (DDP) was used for scalability. The policy was updated following the clipped surrogate objective with value and entropy regularization:

$$\mathcal{L}_{\mathrm{PPO}} = \mathbb{E}_t\Big[ \min\big(r_t(\theta)\hat{A}_t, \ \mathrm{clip}(r_t(\theta), 1 - \epsilon, 1 + \epsilon)\hat{A}_t\big) - c_v(V_\theta(s_t) - R_t)^2 + c_e\, \mathcal{H}[\pi_\theta(\cdot|s_t)]\Big] \quad (7)$$

Where $r_t(\theta) = \frac{\pi_\theta(a_t|s_t)}{\pi_{\theta_{\mathrm{old}}}(a_t|s_t)}$ is the probability ratio between new and old policies, $\hat{A}_t$ is the GAE advantage, $R_t$ is the return, and $\mathcal{H}$ is the entropy bonus. This stage allowed the model to refine beyond demonstration data and adapt to task-specific dynamics, while retaining the inductive bias imposed by the connectome architecture.

## 4 EXPERIMENTS

We evaluated flyGNN on four locomotor tasks: gait initiation, straight walking, turning, and flight within the flybody physics simulator (Vaxenburg et al., 2025). The environments followed the default flybody setup, with binocular visual signals added to the original sensory inputs in the walking tasks. This embodied setting provided diverse multimodal feedback, including proprioception, mechanosensation, and vision, enabling us to test whether a connectome-structured network could flexibly generate stable control policies across different modes of movement. Detailed training process and parameters are provided in the Appendix A and detailed observation and action definitions are provided in Appendix B. The demonstration videos for each task are available on our project webpage.

The flyGNN controller was evaluated under two primary task configurations. For walking-related tasks (gait initiation, straight walking, and turning), the model processed a high-dimensional observation space of 1,253 features, integrating proprioceptive, exteroceptive, and augmented visual inputs. It subsequently output continuous actions of 59 dimensions to precisely control joint actuators and adhesion mechanisms. For the flight task, an observation space of 104 features was implemented, focusing on the core proprioceptive and kinematic states, which generated 12-dimensional actions to modulate wing torques, body joints and a wing pattern generator. The training process for both configurations adhered to the pipeline mentioned above.

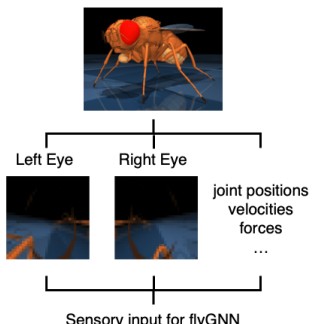

Figure 3: **Demonstration of sensory input for locomotion tasks.** Visual inputs from both eyes are integrated with other sensations to form the sensory input.

**Gait Initiation**

Building on the training pipeline described above, we evaluated flyGNN on a walking task at a target velocity of 3 cm/s. We first examined the process of gait initiation, focusing on the transition from rest to the onset of stable locomotion. Figure 4 illustrates snapshots of the simulated fly prior to the first complete gait cycle. The initiation phase lasted for roughly the first 80 ms, during which irregular steps gradually gave way to rhythmic and coordinated leg movements.

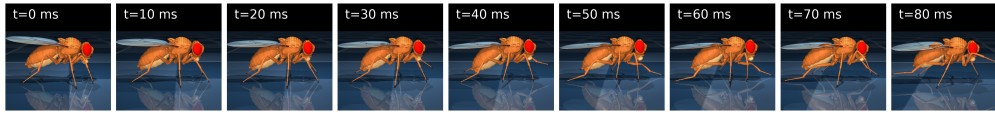

Figure 4: **Gait initiation dynamics.** Snapshots of the simulated fly during the onset of locomotion, prior to the first complete gait cycle. The sequence illustrates how the agent transitions from rest into stepping, with irregular and asymmetric leg movements gradually giving rise to a coordinated pattern.

**Straight-Line Walking**

Following the analysis of gait initiation, we next evaluated flyGNN on a straight-line walking task at a target velocity of 3 cm/s. As illustrated in Figure 5, the model produced stable forward locomotion with clear tripod coordination. The simulated fly maintained a consistent body trajectory over hundreds of milliseconds, without exhibiting drift or collapse, indicating that the learned controller generalizes well to sustained walking.

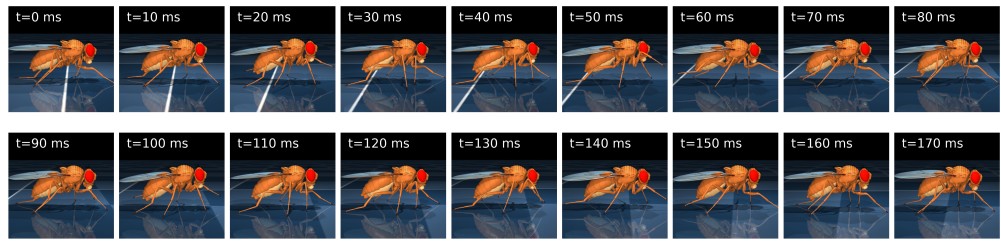

Figure 5: **Walking dynamics.** Snapshots of the simulated fly walking in a straight line at a velocity of 3 cm/s. The model maintains stable stepping sequences with tripod coordination emerging naturally from flyGNN policy model.

Joint-level analysis in Figure 6 shows that actuator outputs are tightly coupled with kinematic trajectories: contralateral legs alternate in phase, producing the classical tripod gait pattern seen in Drosophila. These results demonstrate that flyGNN is sufficient to generate stable straight walking once locomotion is initiated.

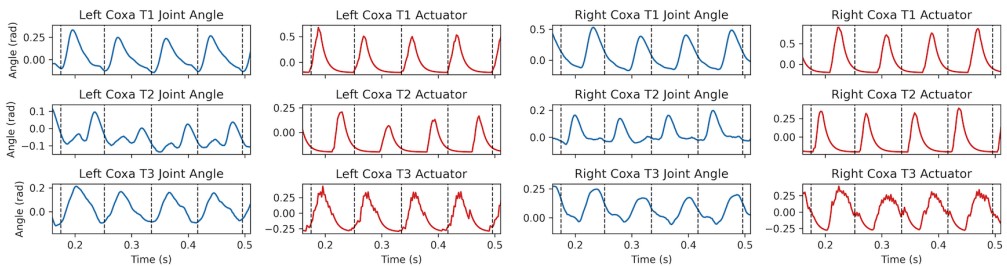

Figure 6: **Joint kinematics and actuator activations during walking.** We visualize the angles (blue) and actuator activations (red) of coxa joints from left (T1–T3) and right (T1–T3) legs over multiple gait cycles. Dashed vertical lines mark gait phases based on the troughs of the left T1 coxa. The model reproduces alternating tripod-like coordination, with left T1/T3 synchronized with right T2, and left T2 synchronized with right T1/T3, consistent with expected fly walking patterns.

**Turning**

We next assessed whether the same policy could generalize to directional maneuvers. In the turning task, the model was instructed to walk at a forward velocity of 3 cm/s while executing a leftward turn at 10 rad/s. As shown in Figure 7, the simulated fly successfully produced a smooth curved trajectory by modulating stride lengths asymmetrically across the body: legs on the inner side of the turn reduced their stance amplitude, while contralateral legs extended their strides. This modulation of gait symmetry arises naturally from the network dynamics, without requiring task-specific tuning or additional control rules.

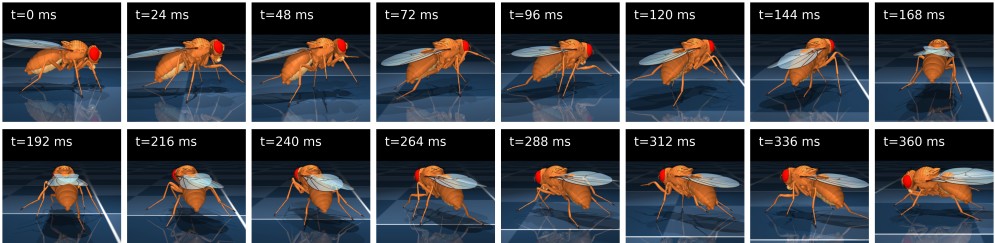

Figure 7: **Turning dynamics.** Snapshots of the virtual fly executing a high-speed left turn at 3 cm/s and 10 rad/s. Over the course of the trajectory, stride lengths on the turning side decrease while those on the contralateral side increase, producing a smooth curved trajectory. This asymmetry demonstrates that flyGNN can generalize beyond straight walking to produce directed maneuvers.

The ability to perform both straight walking and turning indicates that the learned flyGNN policy does not simply memorize a single stereotyped gait, but rather encodes a flexible control strategy that can adapt to new locomotor demands. These findings highlight the robustness of the architecture and suggest its potential for modeling a wider repertoire of multi-task behaviors.

**Flight**

To assess whether flyGNN can generalize beyond terrestrial locomotion, we additionally trained the fly to perform a flying task. In this setting, the policy served as a higher-level neural controller that modulated the output of the wing-beat pattern generator, thereby enabling stable flight dynamics.

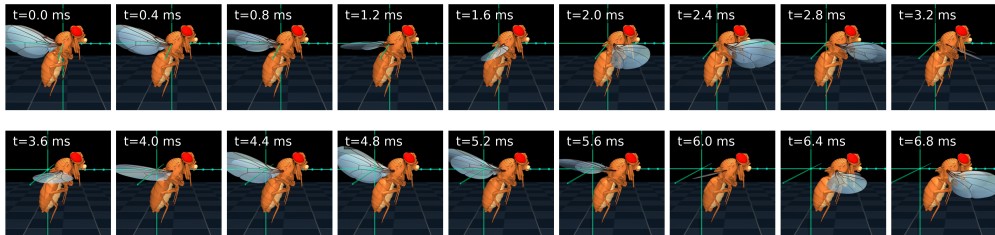

Figure 8: **Flight dynamics.** Snapshots of the virtual fly executing a straight flight task at a velocity of 20 cm/s.

As shown in Figure 8, the trained controller maintained stable forward flight at a constant speed and kept body orientation aligned with the target direction, demonstrating that the connectome-based network can extend from walking to flight locomotion.

These results suggest that connectome-structured networks are not limited to walking but can support a broader repertoire of multimodal behaviors, highlighting the generality of this modeling framework for embodied control.

**Neural Representations Analysis**

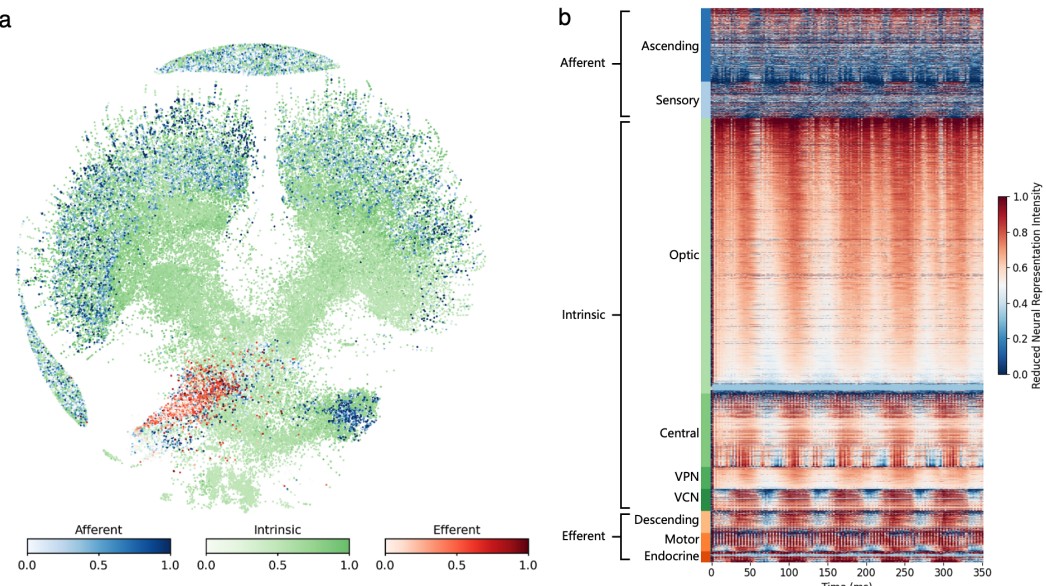

Figure 9: **Neural representations during turning task (0–350 ms). (a)** Force-directed graph layout of the fly connectome at 200 ms, with neurons colored by flow annotations (afferent: blue, intrinsic: green, efferent: red) and reduced neural representation intensity. **(b)** Temporal dynamics of reduced neural representation intensity from 0–350 ms, grouped by flow and superclass annotations. Each row corresponds to one neuron, sorted within its superclass by similarity, and group widths are schematic rather than scaled to neuron counts. The visualization reveals rhythmic patterns aligned with the gait cycle and distinct functional differentiation across superclasses. These results indicate that functional specialization of neurons emerges from the connectome during locomotor control.

One of the key motivations for implementing locomotion through a connectome-based architecture is the opportunity to analyze internal neural representations and study how information propagates through the network. By examining the activity patterns of individual graph nodes, which correspond to neurons, we can ask whether structured and meaningful dynamics emerge, thereby offering a form of connectome-level interpretability. To probe this question, we rolled out trained flyGNN model in simulation and recorded the representations of all neurons at every timestep. These high-dimensional representations were reduced and normalized, yielding the quantity we term reduced neural representation intensity, a proxy for neural activity similar to measures commonly used in neuroscience.

We visualized these reduced neural representations in two complementary ways. First, force-directed graph layouts were used to highlight how intensity distributes across the connectome topology. As shown in Figure 9(a), the snapshot at 200 ms of a turning task reveals structured spatial patterns. When paired with synchronized video of locomotion (available on our anonymous project page), these dynamics reveal that subsets of neurons exhibit rhythmic and heterogeneous oscillations distributed across superclasses. This suggests that the graph's structure itself induces meaningful partitioning of activity rhythms.

Second, we adopted visualization which aggregates representations by annotated flow and superclass labels. We employed random down-sampling to balance visibility across groups. Within each superclass, neurons were reordered using a spectral method based on the Fiedler vector of the Laplacian constructed from similarity matrices of neural activity, thereby clustering neurons with similar dynamics. Despite the fact that our model only encodes neurons by flow type rather than by finer superclass distinctions, the result shown in Figure 9(b) highlights distinct temporal organization across superclasses. The figure demonstrate that reduced neural representation intensities not only align with locomotor rhythms but also uncover emergent specialization across superclasses, showing how connectomic topology alone drives differentiated functional roles. Similar patterns were also

observed in straight-walking tasks (see Appendix E), indicating that these findings are consistent across episodes and tasks.

Taken together, these findings demonstrate that flyGNN develops functional specialization among neuronal groups directly from connectomic structure, without explicit supervision. The emergence of distinct, task-aligned activity profiles suggests that the architecture not only generates behaviorally valid locomotion but also yields potentially interpretable internal representations.

## 5 CONCLUSION AND DISCUSSION

This study demonstrates that the fly-connectomic Graph Neural Network can serve as a reinforcement learning controller for the high-dimensional dynamical systems of a simulated fruit fly to achieve diverse movement tasks such as gait initiation, straight walking and turning. By structuring information flow according to the FlyWire connectome, flyGNN replaces fully connected policy networks with a graph-based network that directly reflect the wiring diagram of the brain.

These results demonstrate that structural priors at the connectome scale provide a powerful inductive bias for embodied reinforcement learning. Even when simplified to an unweighted directed graph without synapse counts or neurotransmitter types, the connectome is sufficient to drive diverse high-dimensional motor control. This suggests that wiring diagrams, long viewed as static anatomical maps, can be directly instantiated as functional networks for closed-loop control.

From the perspective of machine learning, flyGNN offers a new disign paradigm for neural architecture design. Instead of relying on existing artificial structures, we can adapt structure from evolved biological networks. This provides a systematic alternative to generic architectures, potentially improving data efficiency, stability, and transferability across tasks. The approach also opens the possibility of scaling to larger connectomes and more complex agents, where the inductive biases of real nervous systems may be particularly advantageous.

Several limitations highlight opportunities for future progress. First, our implementation simplified biophysical details, leading to the loss of important biological information. Incorporating richer information could improve biological fidelity. Second, compared to MLP-based controllers, our model requires longer per-step computation and higher memory usage, which could be improved. Finally, extending the framework beyond locomotion will provide a more comprehensive test of the generality of connectome-based control.

In summary, flyGNN demonstrates that whole-brain connectomes can be used for embodied motor control. By grounding policy architectures in biological wiring diagrams, this approach suggests a possible towards more human-aligned AI systems, where the inductive biases that shape adaptive behavior in animals can be systematically transferred to artificial agents.

ETHICS STATEMENT

This work does not involve human or animal subjects, private or sensitive data, or potentially harmful applications. The research uses publicly available connectome datasets and simulated environments, and follows the ICLR Code of Ethics in ensuring fairness, transparency, and responsible stewardship of trustworthy research.

REPRODUCIBILITY STATEMENT

We provide detailed model descriptions and training procedures in the main text, with implementation details in the appendix. Methods for visualizations are described, and source code is available at our anonymous webpage.

LARGE LANGUAGE MODELS USAGE STATEMENT

We acknowledge the use of Large Language Models (LLMs) to assist with improving the grammar and clarity of the manuscript. The authors carefully reviewed and verified all content to ensure accuracy and correctness. The authors take full responsibility for the final version of this work.

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

## A  DETAILS OF MODEL ARCHITECTURE AND TRAINING PROCESS

### A.1  MODEL ARCHITECTURE AND IMPLEMENTATION

The flyGNN model is implemented using the PyTorch Geometric (PyG) (Fey & Lenssen, 2019; Fey et al., 2025) library to construct a graph neural network that directly mirrors the Drosophila connectome. The network architecture is structured as a directed graph where nodes represent neurons and edges represent synaptic connections, partitioned into afferent, intrinsic, and efferent sets based on information flow. The model processes inputs through the following components:

Input Normalization: A RunningNorm layer is applied to observations to stabilize training by maintaining running estimates of mean and variance. This layer operates online and is updated during training.

Encoder: Observations are projected into the afferent neuron state space via a linear layer followed by ReLU activation.

Graph Propagation: Information is propagated through the connectome using a message-passing mechanism with gated updates combining previous states and incoming messages.

Decoder: Efferent neuron states are aggregated and passed through a multi-layer perceptron with ReLU activations, outputting action means and standard deviations via linear and softplus heads, respectively.

The model uses simplified biological assumptions—connections are treated as unweighted and directed, ignoring synaptic counts or neurotransmitter types. All experiments were conducted on servers equipped with NVIDIA A100 80GB PCIe GPUs and Intel Xeon Gold 6348 CPUs.

### A.2  TRAINING PIPELINE

Training proceeds in two stages:

Imitation Learning: The policy is initialized by mimicking expert trajectories generated by a pretrained MLP controller provided by flybody. This stage uses PyTorch Lightning for distributed training across multiple workers.

Reinforcement Learning: The model is fine-tuned with Proximal Policy Optimization to maximize task rewards. This stage is implemented using PyTorch Distributed Data Parallel (DDP) for scalability.

Optimization uses AdamW with a learning rate scheduler (ReduceLROnPlateau) that reduces the learning rate upon validation loss improving. Both stages employ gradient clipping to ensure stability.

### A.3  TASK-SPECIFIC CONFIGURATIONS

Walking Task:

- Observation dimension: 1,253 (proprioceptive + exteroceptive + visual inputs),
- Action dimension: 59 (joint actuators and adhesion controls),
- Learning rate: $1 \times 10^{-4}$,
- Message-Passing: GraphSAGE, 16 node channels, 4 message-passing layers,
- Expert policy architecture: 512–512–512–512 fully-connected layers with LayerNorm + Tanh (first layer) and ELU activations, outputting mean (Linear) and std (Linear + Softplus) for 59-dimensional actions.

Flight Task:

- Observation dimension: 104 (simplified proprioceptive and kinematic inputs),
- Action dimension: 12 (wing torques, pattern generator modulation, body joints),
- Learning rate: $1 \times 10^{-5}$,

- Message-Passing: GraphSAGE, 8 node channels, 4 message-passing layers,
- Expert policy architecture: 256–256–256 fully-connected layers with LayerNorm + Tanh (first layer) and ELU activations, outputting mean (Linear) and std (Linear + Softplus) for 12-dimensional actions.

Both tasks use identical graph topology derived from the FlyWire connectome but differ in input/output dimensions and network hyperparameters due to behavioral constraints. Training metrics are logged for performance monitoring.

## B  DETAILED SETTINGS OF LOCOMOTION ENVIRONMENT

Our experiments follow the locomotor task design introduced in the flybody simulator, which provides imitation-learning datasets for terrestrial (*walking*) and aerial (*flight*) behaviors.

We evaluate four tasks:

- Gait initiation: generating stable stepping patterns from rest,
- Straight-line walking: tracking forward centre-of-mass (CoM) trajectories,
- Turning: executing lateral turns at constant speed,
- Flight: stabilizing and steering free flight trajectories.

For walking-based tasks, the default setting of flybody receives a 741-dimensional proprioceptive/exteroceptive observation, comprising:

- accelerometer (3), gyro (3), velocimeter (3), world $z$-axis (3),
- actuator activations (59),
- appendage pose (21), force sensors (18),
- joint positions (85) and velocities (85),
- tactile contacts (6),
- reference displacement (195) and root quaternion (260).

We augment this with binocular visual input: left and right eye cameras ($32 \times 32 \times 3$ RGB each) downsampled and resized to two $16 \times 16$ grayscale figures. The final observation dimension is 1,253.

Actions remain 59-dimensional, actuating adhesion, head/abdomen motion, and all leg joints as in default settings of flybody walking task.

For the flight task, we keep the flybody sensory design with a 104-dimensional input comprising:

- accelerometer (3), gyro (3), velocimeter (3), world $z$-axis (3),
- joint positions (25) and velocities (25),
- reference displacement (18) and root quaternion (24).

The policy outputs 12 control signals: instantaneous wing torques, head/abdomen angles, and Wing-Pattern Generator (WPG) frequency modulation. The WPG provides a nominal wing-beat template, while the policy learns residual corrections.

## C  ABLATION STUDIES ON MESSAGE-PASSING OPERATORS

We evaluated flyGNN across several message-passing operators, including GCN (Kipf & Welling, 2017), EdgeCNN (Wang et al., 2019), GAT (Veličković et al., 2018), GraphSAGE (Hamilton et al., 2018) and PNA (Corso et al., 2020). Table 1 shows the evaluation KL loss and the average reward within 500 episodes of different message-passing operators. We used the imitation learning dataset provided by flybody, and because different imitation trajectories exhibit variability, the rewards naturally show high fluctuation. The results show that simpler operators (e.g., GCN, EdgeCNN) generally converged with higher KL divergence to the teacher and lower average reward, while more

Table 1: Performance comparison of flyGNN with different message-passing operators.

| Model | Node dim | Depth | Eval KL ↓ | Avg. reward ↑ |
|---|---|---|---|---|
| GCN | 16 | 4 | 7.93 | 43.13 |
| GraphSAGE($c = 2$) | 2 | 6 | 6.85 | 62.85 |
| EdgeCNN | 4 | 2 | 5.54 | 90.48 |
| GAT | 8 | 2 | 3.43 | 132.65 |
| GraphSAGE | 16 | 4 | 3.31 | 125.55 |
| PNA | 4 | 2 | 2.89 | **145.33** |

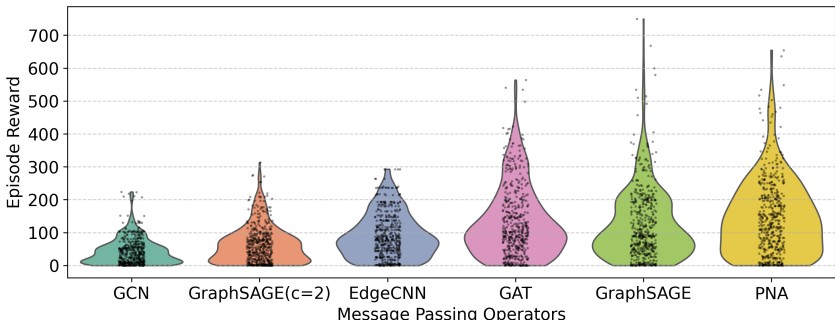

Figure 10: **Performance across message-passing operators.** Distribution of episode rewards obtained by GCN, GraphSAGE, EdgeCNN, GAT, and PNA. More expressive operators (GraphSAGE, PNA) achieved higher average rewards and more stable rollouts compared to simpler operators (GCN, EdgeCNN).

expressive operators (e.g., PNA, GraphSAGE) achieved closer distributional matching and longer average rollout stability.

Across all variants, the models exhibited stable locomotion; notably, even the weaker ones retained sufficient ability for walking and turning, and an extreme GraphSAGE variant with only two node channels still performed acceptably.

## D  NEURAL DYNAMICS DURING GAIT INITIATION

To probe the neural dynamics underlying gait initiation, we visualized internal neuron states sampled from the same episode before the first stable gait cycle. Specifically, we collected 16-dimensional features from all 139,246 neurons at selected times $[0, 20, 40, 60, 80]$ ms and reduced them to two dimensions using UMAP (McInnes et al., 2018).

Figure 11 reveals clear temporal changes in the organization of neural states. At initialization (0 ms), neuron features were randomly distributed, with optic and central neurons largely overlapping. As the episode progressed, neurons formed increasingly distinct clusters, with optic and central populations separating into identifiable regions by 80 ms. This indicates that the flyGNN architecture simulates connectome-based neural dynamics that promote functional specialization of different neural groups during locomotor tasks.

These results suggest that the flyGNN framework can capture meaningful reorganization of neural states during behavior. The observed emergence of functional segregation highlights the potential of this approach for investigating neural plasticity. In particular, it may help elucidate the interplay between developmental trajectories, learning mechanisms, and the structural organization of the connectome.

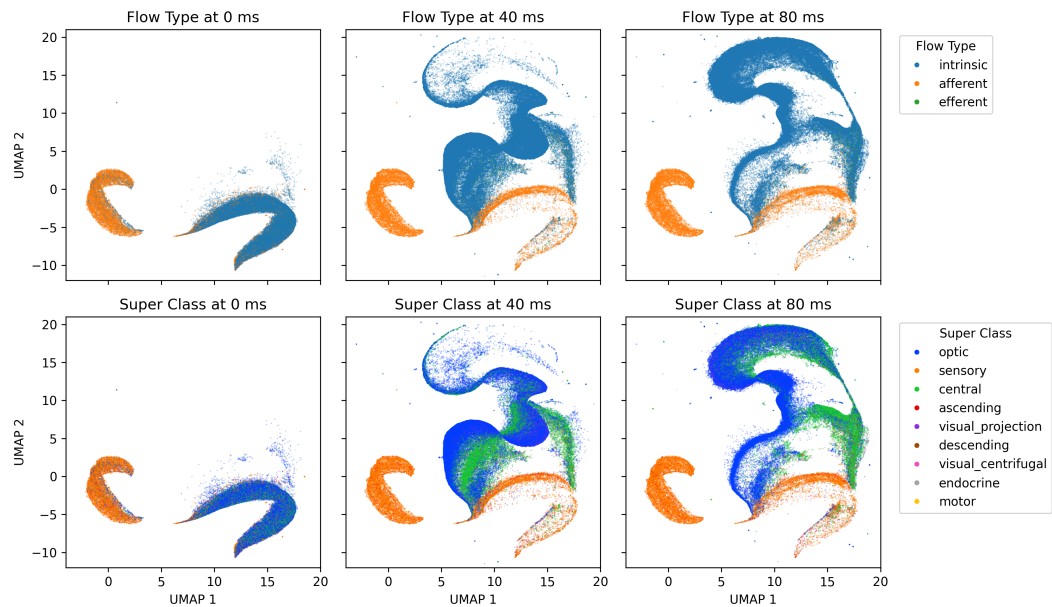

Figure 11: **Neuron state representation during gait initiation.** UMAP projections of neural features at different time points (0 ms, 40 ms, 80 ms). Top: neurons colored by FlyWire flow type (afferent, intrinsic, efferent). Bottom: neurons colored by 'superclass' annotations. The visualization shows that initially mixed features gradually separate into distinct clusters, with optic and central populations becoming clearly differentiated by 80 ms. This structured reorganization indicates that the flyGNN policy induces meaningful neural dynamics, supporting functional specialization as gait coordination emerges.

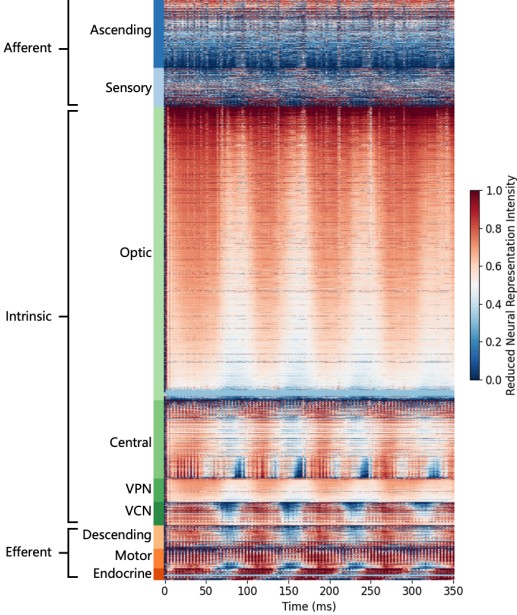

Figure 12: **Neural representations during straight walking (0–350 ms).** Temporal dynamics of reduced neural representations, grouped by flow and superclass annotations. Widths of superclass groups are schematic and not proportional to actual neuron counts. The visualization reveals rhythmic activity aligned with gait cycles, while motor and descending groups show sparser but phase-locked dynamics. Unlike turning, the straight-walking condition lacks the optic phase shift, indicating more symmetric activation patterns during forward locomotion.

# E    VISUALIZATION OF NEURAL REPRESENTATIONS IN WALKING TASK

To complement the turning-task analysis, we also visualized reduced neural representation intensities during walking task (see Sec. 4). Figure 12 shows temporal dynamics of neural activity from 0–350 ms, including the first 80 ms of gait initiation, after dimensionality reduction with principal component analysis and normalization. Neurons are grouped by flow type and superclass annotations, with width adjusted schematically for visibility, and reordered within each superclass using spectral similarity sorting.

The visualization highlights distinct temporal activation profiles across neural groups. In particular, optic and central populations display rhythmic modulations aligned with gait cycles, while motor and descending neurons show sparser but phase-locked activation. Compared to the turning condition, the straight-walking trajectories exhibit more symmetric activation patterns without the optic phase shift observed during turning. This difference suggests that lateralized activation dynamics may specifically support asymmetric motor outputs during directional maneuvers, whereas straight walking is driven by more synchronized, globally coordinated patterns.

# F    DETAILED METHOD FOR VISUALIZATION

## F.1    FIGURE 2(A)

We visualized the aggregated synapse graph of the fly connectome using the FlyWire FAFB v783 dataset, with nodes classified by superclass labels provided by FlyWire. The node size reflects the number of neurons in each class, while the thickness and darkness of the directed edges represent the number of aggregated synaptic connections.

## F.2    FIGURE 2(B)

In the visualization of the network of connectome, we did not incorporate the three-dimensional structural priors of Drosophila neurons. Instead, we employed a force-directed layout algorithm (Kobourov, 2012) for graph drawing. In this method, edges are modeled as springs while nodes exert repulsive forces on each other, and the system iteratively evolves toward a low-energy equilibrium. This layout highlights the topological organization of the network, making the relationships among different super-classes and subgroups more visually apparent. For visualization, we applied a threshold of more than 25 synapses and retained only connections exceeding this cutoff.

## F.3    FIGURE 9(B)

Figure 9(b) presents the temporal dynamics of reduced neural representation intensity across the fly connectome during a 350 ms turning maneuver. To visualize population activity patterns, we first extracted high-dimensional neural features from the model, with shape $T \times N \times C$ (time × neurons × channels). These features were compressed into a single-channel intensity measure via PCA, retaining only the first principal component along the channel dimension. The resulting values were then clipped to the 5th–95th percentile range and min-max normalized to $[0, 1]$ to improve comparability and reduce outlier effects.

Due to the extreme imbalance in superclass sizes (e.g., $> 77000$ neurons in optic classes versus only 106 in motor classes), we employed stratified random downsampling to ensure visibility across all functional groups while preserving relative proportions. The downsampling thresholds were set as follows: sensory: 400, ascending: 200, optic: 1500, central: 400, visual projection: 120, visual centrifugal: 120, descending: 120, motor: 100, endocrine: 60. This approach maintained the diversity of neural responses while creating visually interpretable group representations.

