# OpenReview forum: "Whole-Brain Connectomic Graph Neural Network Enables Whole-Body Locomotion Control in Drosophila"
_ICLR.cc/2026/Conference — Submitted to ICLR 2026_

### Official Review · Reviewer_2soe · 2025-10-29

**Soundness:** 2
**Presentation:** 4
**Contribution:** 3
**Rating:** 6
**Confidence:** 3

**Summary:**

This paper stands at the intersection of embodied AI and neuroscience, and introduces a novel paradigm which leverages whole-brain connectome for designing learning-based control policies. The proposed flyGNN that mirrors Drosophila connectome is validated on a biomechanical model of fruit fly in MuJoCo and shown to successfully realize four locomotion tasks, spanning gait initiation, walking, turning and flight. The paper also provides in-depth analysis of dynamic neural activities and specialization within flyGNN, yielding meaningful and interpretable results that could also inspire neurobiology studies.

**Strengths:**

1.This paper addresses an interesting yet under-explored research problem, i.e., embedding whole-brain connectome into controller architectures to yield biologically inspired computational models. It displays a promising avenue for bridging neuroscience and artificial intelligence and how they could provide valuable insights for each other.

2.The proposed fly-connectomic Graph Neural Network (flyGNN) is validated within the flybody physics simulator across four locomotion tasks, showing successful task completion.

3.The abundant visualization results reveal the potential of flyGNN to discover dynamic neural activity patterns, as well as uncover emergent specialization across different functional roles.

**Weaknesses:**

1.The paper lacks any performance comparisons with simpler neural architectures, such as MLPs and vanilla graph neural networks. This leaves it questionable whether the introduction of connectomic inductive biases truly benefits the learning of intelligent behaviors.

2.The authors are suggested to describe in greater detail the training procedure of flyGNN. For example, why do they adopt an imitation learning phase, in contrast to many other robotic control policies that are trained using RL from scratch. Moreover, it is not clearly explained how the MLP-based policy was originally trained, including the source of expert demonstrations.

**Questions:**

1.Could the authors describe the labors required for converting the original FlyWire model into the code of the controller architecture? Is any automation feasible during this process?

2.Following Weaknesses 1, given the higher computation and memory budges of flyGNN, have the authors validated the actual performance gains over simple architectures? Specifically, could the authors provide quantitative comparisons between flyGNN and MLP/vanilla GNNs with comparable parameter amounts?

---

> ### Author Response · Authors · 2025-12-03
> **Response to Reviewer 2soe**
>
> Thank you for highlighting the novelty of our work.
>
> **Q: Automation Process of FlyWire.**
>
> **A:** The conversion process is largely automated. Once the FlyWire dataset is downloaded from Codex, our pipeline directly parses neuron IDs, synaptic edges, and flow-type annotations, constructs the directed adjacency matrix, and aligns afferent/efferent neurons with the simulator’s observation and action interfaces. No manual reconstruction or hand-editing of the connectome is required. The same pipeline can be applied to other connectome datasets with similar formats.
>
> **Q: Computational Cost and Performance Comparisons.**
>
> We acknowledge the importance of benchmarking the connectome-based model against systematically designed comparison models. We will include these quantitative comparisons in the future version.

---

### Official Review · Reviewer_Vmi7 · 2025-10-30

**Soundness:** 1
**Presentation:** 4
**Contribution:** 2
**Rating:** 2
**Confidence:** 4

**Summary:**

The authors propose FlyGNN: a graph neural network which uses the Drosophila connectome as a template for its connectivity. The authors claim to simulate the full neural network of the Drosohpila, which numbers upwards of 100,000 neurons. The show that such a network can be trained to control a MuJoCo version of Drosophila that can both walk and fly.

**Strengths:**

The paper is very well presented and the results compelling from an engineering standpoint. Getting this to work must have required a substantial amount of work given the complexity of the model. However, the authors show convincing results that the Drosophila model learns to locomote.

**Weaknesses:**

The authors stated motivation is to bridge the gap between connectomics and neural network circuits that can learn (in their own words t answer the question: "how can static connectomes be transformed into dynamic, functional models that reproduce the intricate and adaptive motor behaviors of animals?". This is all well and good, but to what end? What questions does this merging between connectomics and "plastic" neural networks enables? What can we learn about Drosophila or about the relation between connectomics and neural networks more broadly? We can't say because the authors don't elaborate on any of it.

**Questions:**

Here are some question that I believe should be answered before this work can be accepted:

1. What is the benefit of including this connectomic information?
2. Does the simulated Drosophila learn faster for example or is more robust?
3. What are sensible null models to compare against? What do they tell us about the influence of a powerful model such as the GNN.
4. Is the GNN variant that the authors settle on more or less biologically plausible? Is it not contradictory with the stated goal of adding more biological realism?

---

> ### Author Response · Authors · 2025-12-03
> **Response to Reviewer Vmi7**
>
> We thank the reviewer for the thoughtful comments. Below we address the key concerns.
>
> **Q: Scientific motivation and significance**
>
> **A:** We agree that our motivation can be stated more explicitly. To our knowledge, this work is among the first attempts to instantiate the adult Drosophila connectome as a closed-loop controller capable of producing stable locomotor behaviors. This offers a computational framework for examining how whole-brain connectivity may relate to embodied function and behavior.
>
> **Q: Null models and comparisons.**
>
> **A:** We fully agree that comparing against baselines and null models is essential. We will include these comparisons in the future version.
>
> **Q: Biological plausibility.**
>
> **A:** Our focus is on understanding how the connectome’s structural topology supports information propagation and task execution. Ablation results on different message-passing mechanisms show that the specific operator affects performance moderately but does not change the overall behavior. We therefore use an abstract GNN formulation that is trainable and scalable for locomotion control. Achieving full biophysical realism is not the primary aim of our work, as such models are currently challenging to optimize and may not be well suited for high-dimensional embodied tasks.

---

### Official Review · Reviewer_DbnM · 2025-10-31

**Soundness:** 2
**Presentation:** 3
**Contribution:** 2
**Rating:** 2
**Confidence:** 5

**Summary:**

The paper expands upon Vaxenburg et al., 2025 with a policy network constrained by the unweighted, directed structure implied by the Flywire Drosophila connectome dataset. The authors separate the network into afferent, intrinsic, and efferent neurons then decode efferent activity to continuous motor commands. The advertised contribution is a connectome-structured GNN that can drive walking, turning, and flight with the same architectural class. The environments, embodiment, and demonstration data derive from the Vaxenburg paper. The authors do not present a head-to-head comparison, they demonstrate that the connectome architecture can also achieve stable behavior. The authors claim that the connectome provides an inductive bias which yields interpretable internal organization. To support this claim, they present reduced-dimensional node activations which at times show rhythmic, compartment-specific structure during behavior and emergent segregation across superclasses. To train the network, the authors train the parameters of a policy whose topology is fixed by the unweighted brain graph first by imitation of the MLP "expert" and then by PPO fine-tuning. The trained weights are the encoder into the afferents, the afferent gate, the message passing operator on the graph, and the readout from efferents (and a separate MLP value network for PPO)

**Strengths:**

The paper is ambitious in scope, incorporating the entire Drosophila brain, and the authors make an important stride towards understanding this connectome in an embodied setting by replicating the results of the Vaxenburg paper. The authors also successfully demonstrate that a network structured like the connectome is capable of supporting all behaviors simultaneously. The implementation of the network as a GNN is a sound way to deal with the recurrence of the connectome and a source of novelty over the Vaxenburg paper.

**Weaknesses:**

The authors omit details which are thought within the field to be very important for the working of the connectome- most importantly, this includes the number of synapses between neurons and the neurotransmitter, or at least the sign of the neurotransmitter. Especially lacking a threshold on number of synapses, the unweighted graph the authors use is probably too noisy and lacks the most important pathways (generally, the strongest connections a neuron makes have orders of magnitude more synapses than the weakest, and these weak connections are likely due to both developmental and reconstruction noise see: Flywire Schlegel paper). Additionally, the relevance of the rhythmic activity that the authors claim is thin given the lack of excitatory and inhibitory connections and the lack of a ventral nerve cord, which is putatively thought to be the primary driver of rhythmic activity in the motor tasks the authors describe (chiefly locomotion see: work of Eve Marder and John Tuthill for foundational and new, Drosophila-specific work respectively). From this newer work, it is known that descending neurons mostly encode low-frequency decisions, and the neural enactment of these decisions towards coordinating patterns of motor activity occurs in the VNC, which would be important to model. This does not preclude the importance of rhythms in the central brain, but the authors should further describe the neurons in which rhythms are present to discuss the biological plausibility or importance of these connections. The authors present a surrogate for neural activity in their neural representation intensity, but I would expect either a novel prediction (ie: specific neurons which are correlated with a specific behavior or characteristic of that behavior) and/or a connection to literature (an extrinsic model validation based on the numerous papers connecting central brain neurons to specific characteristics of behavior).
On the machine learning side, I question whether the results here support the claim of an inductive bias. I would expect any trained network to be able to learn function to some epsilon (Cybenko); the fact that a network can be trained to some task is not necessarily proof of a beneficial inductive bias. Additionally, claims of an inductive bias are somewhat confounded by the choice to distill from the expert MLP in the first place. To prove this claim, the authors should have had some baseline, for instance a dense network with the same number of neurons, a sparse network with the same number of neurons and edges, and/or a shuffling of the connectome's edges, maintaining the afferent/intrinsic/efferent partitions.

**Questions:**

The following are a superset of questions that if answered would make the paper stronger:
1) You treat the connectome as an unweighted- can you report sensitivity analysis where you (i) weight edges by synapse counts; (ii) threshold the graph by min-synapses; and (iii) incorporate sign (even heuristically from transmitter labels where available)?
2) Towards inductive bias, could you aadd baselines? (i) a dense MLP matched on parameter count and inference time; (ii) a sparse random graph matched on nodes/edges; (iii) a degree-preserving edge rewire that maintains afferent/intrinsic/efferent partitions. Report sample efficiency, asymptotic return, and robustness under sensor noise.
3) From the MLP teacher-student, can you prove the GNN shows advantages beyond simple distillation? Could you report (i) head-to-head returns vs. the MLP teacher on identical seeds; (ii) zero-shot generalization where the student outperforms the teacher under perturbations; and (iii) learning curves from scratch (no IL) to test whether the topology aids exploration/credit assignment.
4) You mention the increased computational expense of the GNN model - could you report some quantification there? These could include parameter counts, FLOPs, and wall-clock inference time for flyGNN vs. MLP teacher.
5) Much of the evidence is qualitative (snapshots/plots). Could you provide standardized metrics per task (e.g., CoM tracking error, fall rate, gait-phase stability, flight speed variance), and show confidence intervals over seeds?

---

> ### Author Response · Authors · 2025-12-03
> **Response to Reviewer DanM**
>
> Thank you for reviewing our paper and for your valuable feedback.
>
> **Q: Sensitivity Analysis on Synapse Count and Threshold**.
>
> **A:** Thank you for your suggestion. We will incorporate considerations of synapse counts and transmitter labels, as well as corresponding sensitivity analyses in the future version of this work.
>
> Regarding the threshold issue you raised, we utilized the **Connections (Filtered)** dataset from Codex, which uses 5+ synapse threshold. This dataset has been widely adopted in several FlyWire-related studies for data analysis. We believe that adhering to this dataset ensures consistency between our work and FlyWire.
>
> We have carefully reviewed the FlyWire Schlegel paper and noted the point you mentioned that *"weak connections are likely due to both developmental and reconstruction noise."* While the paper states that *"2.6% edge weight (or 31 synapses) can be considered to be strong,"* it is important to highlight that the definition of “edge” in that context refers to connections between cell types, rather than connections between individual neurons. As such, the edge counts reported in Schlegel et al. are inherently aggregated and therefore larger in scale.
>
> Given this distinction, we believe the 5+ synapse threshold remains a reasonable choice. Nonetheless, we plan to explore the effects of using higher thresholds in our future sensitivity analyses.
>
> **Q: Baselines & Standardized Metrics.**
>
> **A:** We fully agree that comparing against baselines and calculating standardized metrics is essential. We will include these quantitative comparisons in the future version.
>
> **Q: MLP teacher-student**
>
> **A:** We conducted a head-to-head comparison between the GNN student model and the MLP teacher under a variety of task conditions. This indicates that the student flyGNN model successfully reproduces its behavior robustly across diverse scenarios.
>
> Our focus is not to claim superiority over the MLP baseline, but rather to highlight that our structured GNN model is **capable of achieving the same level of performance** while being graph-structured and biologically grounded. As such, we believe that performance comparison alone is not the most informative indicator in this case, since both policies already solve the task to near perfection.
>
> Each column corresponds to a task setting: (speed cm/s, yaw_speed rad/s)
>
> | Model                             | speed=2,yaw=0     | speed=3,yaw=0     | speed=4,yaw=0     | speed=5,yaw=0     | speed=3,yaw=4     | speed=3,yaw=7     |
> | --------------------------------- | ----------------- | ----------------- | ----------------- | ----------------- | ----------------- | ----------------- |
> | **flyGNN – Pos Err (mean±std)**   | 0.02334 ± 0.00056 | 0.02578 ± 0.00071 | 0.03114 ± 0.00050 | 0.06622 ± 0.02341 | 0.02263 ± 0.00067 | 0.02351 ± 0.00227 |
> | **MLP – Pos Err (mean±std)**      | 0.02750 ± 0.00051 | 0.03217 ± 0.00120 | 0.03600 ± 0.00089 | 0.05373 ± 0.02184 | 0.02695 ± 0.00054 | 0.02790 ± 0.00081 |
> | **flyGNN – Angle Err (mean±std)** | 4.097 ± 0.063     | 5.802 ± 0.116     | 7.334 ± 0.113     | 13.873 ± 1.753    | 5.619 ± 0.108     | 10.888 ± 1.272    |
> | **MLP – Angle Err (mean±std)**    | 4.519 ± 0.136     | 6.064 ± 0.096     | 7.352 ± 0.127     | 13.589 ± 4.560    | 5.808 ± 0.099     | 8.484 ± 0.162     |

---

### Official Review · Reviewer_7bYN · 2025-11-01

**Soundness:** 2
**Presentation:** 3
**Contribution:** 2
**Rating:** 2
**Confidence:** 3

**Summary:**

This paper introduces the "flyGNN," a novel reinforcement learning (RL) controller for a simulated Drosophila agent. The key innovation is that the network architecture is not a generic multilayer perceptron (MLP) but is directly instantiated from the adult Drosophila whole-brain connectome (FlyWire). The connectome is modeled as a directed graph, partitioned into afferent (sensory input), intrinsic (internal processing), and efferent (motor output) pathways. The authors demonstrate that a single, untrained flyGNN policy, when trained with RL, can successfully control a sophisticated, physics-based biomechanical model ("flybody") to perform diverse locomotion tasks, including gait initiation, straight walking, turning, and flight. The authors claim that this demonstrates the sufficiency of connectome-level structural priors for complex motor control and that this structure leads to the emergence of functional specialization within the network without explicit supervision.

**Strengths:**

- The central idea of using a complete, synaptic-resolution whole-brain connectome as the literal architecture for an RL controller is highly novel. It represents a significant and ambitious conceptual bridge between systems neuroscience and embodied artificial intelligence.

- The ability to train a single policy on this complex, biologically-grounded architecture to perform a variety of distinct locomotion tasks (walking, turning, and even generalizing to flight) is a strong proof of concept. The (implied) videos are a compelling demonstration that the approach is viable.

- By grounding the controller in a known anatomical blueprint, this work opens up a promising new avenue for interpretability. The analysis in Figure 9, while preliminary, shows the potential for linking learned neural dynamics directly to specific, annotated neural classes and potential pathways.

**Weaknesses:**

While the core idea is strong and the results are visually impressive, the paper is currently in a preliminary state and suffers from significant weaknesses in its evaluation, justification of claims, and engagement with existing literature.


- The paper's greatest weakness is the complete absence of quantitative baselines. The authors mention that standard RL policies use "generic multilayer perceptrons" but provide no comparison. To validate the claim that the connectome structure provides a "powerful inductive bias," it is essential to compare flyGNN against standard architectures.
    - Missing Baselines: The paper needs to include comparisons against, at a minimum (1) A standard MLP controller with a similar number of trainable parameters. (2) A generic Graph Neural Network (GNN) with a similar parameter count but on a non-biological graph (e.g., a random graph, a sparse graph) to show that the specific topology of the connectome matters.

- The results are presented descriptively (e.g., "successfully produced a smooth curved trajectory"). There are no quantitative metrics for performance. How fast is the walking? How stable is the gait? What is the tracking error for the turning task? Without these metrics, it is impossible to "judge the advancement" or determine if the controller is performing well or just minimally functional.

- The paper operates at the intersection of several massive fields (connectomics, computational neuroscience, GNNs, RL for locomotion) yet cites remarkably little literature. The "Related Work" section is sparse. Section 3.1 includes no citations, giving the false impression that this modeling approach exists in a vacuum. The discussion section contains no citations at all. The authors make sweeping claims about "a new design paradigm" and "advancing both neuroscience and embodied artificial intelligence" without situating their work relative to any existing studies, such as other neuro-inspired architectures or GNNs in RL, or the vast literature of RL for motor control outside of neuroscience.


- Several key parts of the analysis are described in a vague, unscientific manner. The description of the neural analysis is unclear: "...dimensional representations were reduced and normalized, yielding the quantity we term reduced neural representation intensity... a proxy for neural activity..." This is insufficient. What dimensionality reduction method was used (PCA, UMAP, etc.)? How was it normalized? Why is this a valid "proxy" for neural activity? The clustering method is also opaque: "We employed random down-sampling... neurons were reordered using a spectral method based on the Fiedler vector of the Laplacian constructed from similarity matrices of neural activity..." This is a critical step for their "functional specialization" claim, but it is poorly explained and justified. How was the similarity matrix defined? Why was this specific clustering method chosen?


- The paper's central scientific claim is that "flyGNN develops functional specialization... directly from connectomic structure, without explicit supervision." This claim is not well-supported because it lacks a proper null model. The network is initialized with a highly specific, non-random structure where afferent, intrinsic, and efferent nodes already have vastly different topological roles. It is almost expected that nodes with different structural properties (e.g., inputs vs. outputs) will develop different functional representations during training. The authors fail to disentangle whether the observed specialization is due to the specific FlyWire connectome or just any complex, sparse, graph-structured initialization. A strong null model (e.g., a randomly rewired graph that preserves the in/out-degree distribution) is needed to show that the specific biological wiring is the causal factor.


- The model is complex, incorporating the entire 139k-neuron connectome. The authors state their implementation is "simplified" (unweighted, directed graph). However, they provide no ablation study to identify which components are crucial for success. Is the full-scale connectome necessary? Would a smaller subgraph work? How important is the afferent/intrinsic/efferent partition? Without ablations, it's unclear why the model works, which limits its scientific and engineering insights.

**Questions:**

1. Baselines: Can you provide quantitative performance metrics (e.g., gait stability, speed, task tracking error) and compare them against at least two critical baselines: (a) a standard MLP controller with a similar parameter count and (b) a GNN controller with a non-biological graph?


2. Null Model: To substantiate the claim of "emergent functional specialization," could you provide a comparison against a null model?


3. Please elaborate on the spectral clustering method used in Figure 9b. How was the "similarity matrix" used to construct the Laplacian defined? Please also provide a precise definition for "reduced neural representation intensity"? Specifically, what dimensionality reduction technique was used, and what was the normalization procedure?


4. Ablation Study: Given the model's complexity, what are the most crucial components for its success? Can you provide an ablation study (e.g., simplifying the graph, using a subgraph) to isolate the key drivers of performance?


5. Literature Context: The discussion section makes broad claims but lacks citations. Could you please situate your work within the existing literature on graph neural networks for RL and other neuro-inspired locomotion controllers?

---

> ### Author Response · Authors · 2025-12-03
> **Response to Reviewer 7bYN**
>
> We appreciate your valuable feedback to help us improve our paper. Thank you for recognizing the novelty and ambition of our work.
>
> **Q: Baselines and Null Models.**
>
> **A:** We recognize the importance of systematic baseline evaluations and are preparing controlled comparison studies to address this point. We will include these quantitative comparisons in the future version.
>
> **Q: Details on spectral clustering and neural representation.**
>
> **A:**
>
> (1) Regarding the spectral clustering method in Figure 9b, we employed this technique  to optimize the ordering of neurons within each class **specifically for visualization purposes**.
>
> The similarity matrix ($S$) was defined as a weighted combination of cosine similarity and normalized Euclidean distance. Specifically, for any two neurons $i$ and $j$ with activity vectors $x$, the similarity $S_{ij}$ is calculated as:
>
> $S _ {ij} = \alpha \text{sim} _ {\cos}(x _ i, x _ j) + (1-\alpha) d _ {ij}$
>
> where $\alpha$ balances the two metrics (we used $\alpha=0.7$) and $d _ {ij}$ is the normalized Euclidean distance between $i$ and $j$. Based on this matrix, we computed the symmetric normalized Laplacian ($L = I - D^{-1/2}SD^{-1/2}$) and determined the optimal ordering by sorting the values of the Fiedler vector (the eigenvector corresponding to the second smallest eigenvalue of $L$).
>
> (2) The detailed definition of "reduced neural representation intensity" (PCA-based reduction) is provided in **Appendix F.3**. We first extracted high-dimensional neural features from the model, with shape $T×N×C$ (time × neurons × channels). These features were compressed into a single-channel intensity measure via PCA, retaining only the first principal component along the channel dimension. The resulting values were then clipped to the 5th–95th percentile range and min-max normalized to $[0, 1]$.
>
> **Q: Ablation Study.**
>
> **A:** Currently, we use the whole-brain connectome to respect the biological wholeness. In Appendix C, we explored different message-passing operators. We agree that ablating specific brain regions (subgraphs) is a promising direction to identify minimal functional circuits. We plan to perform these biological ablation studies in future work.
>
> **Q: Literature Context.**
>
> **A:** We appreciate your suggestion to better situate our work within the existing literature.
>
> **1. GNNs in Brain Connectivity:** While there is a growing body of literature applying GNNs to brain connectivity, these works predominantly focus on macro-scale modalities, such as MRI (structural), fMRI/EEG/MEG (functional), or DTI (tractography) [1]. To the best of our knowledge, since the release of the FlyWire dataset, our work is the first to apply GNNs to the **whole-brain connectome at the cellular level** for functional simulation. Unlike the inductive, classification-focused tasks common in neuroimaging, our task leverages the connectome as a native graph structure for a large-scale, transductive, and recurrent control problem.
>
> **2. GNNs in RL and Robotics:**
>
> Our work differs substantially from prior studies that combine Graph Neural Networks (GNNs) with Reinforcement Learning (RL), and as such, we have not found suitable references that closely align with our approach. Most existing GNN+RL research focuses on non-motion control domains, such as knowledge graphs, traffic systems, and modern manufacturing, which are fundamentally different from our application.
>
> As for works involving GNNs, robotics, and RL, we identified several relevant studies. However, due to significant differences in how the graph representations are constructed, these works also exhibit limited relevance to our methodology. Below, we list the representative studies we have reviewed: (1) **Robot Swarms/Multi-Agent Systems:** Modeling interactions between distinct agents (e.g., Tolstaya et al., *Learning Decentralized Controllers for Robot Swarms*). (2) **High-level Planning:** Modeling object manipulation or scene graphs (e.g., Lin et al., *Efficient and Interpretable Robot Manipulation*). In contrast, we utilize GNNs to model the internal *neural* circuit topology rather than external agent interactions or high-level plans. (3) **Physical Morphology Encoding:** GNNs are employed to encode the physical morphology of the agent (e.g., limbs as nodes, joints as edges). These works aim for policy generalization across different physical structures or tasks, often falling under the Multi-Task DRL paradigm. (e.g. Wang et al., *NerveNet: Learning Structured Policy with Graph Neural Networks*)
>
> **3. Neuro-inspired Locomotion Controllers:** We agree that the discussion on neuro-inspired controllers should be broadened. Beyond *NeuroMechFly*, the other significant body of work involves Spiking Neural Networks (SNNs). We will add a discussion comparing our GNN-based connectome approach with existing SNN-based control methods to provide a more comprehensive overview of the field.

---

### Author Response · Authors · 2025-12-03
**General Response**

We thank the reviewers for their insightful and constructive feedback.

We understand the primary concerns regarding quantitative baselines (null models) and the justification for using the connectome. We would like to clarify the primary positioning of this paper:

1. **Feasibility of Connectome-Based Control:** Our main contribution is the demonstration of a neural network instantiated directly from the biological whole-brain connectome of Drosophila can successfully learn to control a complex, physics-based body to perform diverse locomotor tasks. We believe our work is among the first to bridge whole-brain connectomics and embodied control at this scale.
2. **Future Quantitative Validation:** We fully agree with the reviewers that comparisons against MLPs and random graphs are essential to quantify the specific benefit of the connectome topology. As requested, we are actively working on these experiments and will include them in the future version of this work.
3. **Interpretability:** Our goal is to build a platform for aligning simulation with biology. In future work, we plan to validate the emergent neural representations observed in our model against biological experimental data.

---

### Meta-Review · Area_Chair_gMzE · 2026-01-07

**Summary:**

The article studies a reinforcement learning controller whose architecture is instantiated directly from a complete adult Drosophila connectome.

The initial reviews are critical, with concerns about motivation, experimental setup, lack of comparisons. The initial rebuttal acknowledged some of these limitations leaving additional comparisons for future work. I conclude the article advances in an interesting direction but would benefit from revision.

Therefore I am recommending reject.

**Reviewer Concerns:**

Response to Reviewer 7bYN

Q: Baselines and Null Models.
A: We recognize the importance of systematic baseline evaluations and are preparing controlled comparison studies to address this point. We will include these quantitative
comparisons in the future version.

Q: Ablation Study.
A: Currently, we use the whole-brain connectome to respect the biological wholeness. In Appendix C, we explored different message-passing operators. We agree that ablating specific
brain regions (subgraphs) is a promising direction to identify minimal functional circuits. We plan to perform these biological ablation studies in future work.

Response to Reviewer DanM

Q: Baselines & Standardized Metrics.
A: We fully agree that comparing against baselines and calculating standardized metrics is essential. We will include these quantitative comparisons in the future version.

Response to Reviewer Vmi7

Q: Null models and comparisons.
A: We fully agree that comparing against baselines and null models is essential. We will include these comparisons in the future version.

**Reviewer Scores:**

For each review, specify how you think the reviewer would have changed their score if they had been able to participate fully in the discussion.

Reviewer 7bYN: 2 -> 2 / 4
Reviewer DbnM: 2 -> 2 / 4
Reviewer Vmi7: 2 -> 2 / 4
Reviewer 2soe: 6 -> 6

---

### Decision · Program_Chairs · 2026-01-26

Reject